# Socioeconomic markers of dengue mortality in the 100 Million Brazilian Cohort (2007–2018): A nationwide registry-based cohort study

Luciana Lobato Cardim[1]*, Maria da Glória Teixeira[1,2], Maria da Conceição N. Costa[2], Wema Meranda Mitka[3], Camila Silveira Silva Teixeira[1], Andreia C. Santos[3], Gervasio Santos[4], André Portela Fernandes de Souza[5], Liam Smeeth[3], Mauricio L. Barreto[1,2], Elizabeth B. Brickley[3‡], Enny S. Paixao[1,3‡], Julia M. Pescarini[1,3‡]

1 Centro de Integração de Dados e Conhecimentos para Saúde - CIDACS (Center of Data and Knowledge Integration for Health), Instituto Gonçalo Moniz, Oswaldo Cruz Foundation, Salvador, Bahia, Brazil, 2 Instituto de Saúde Coletiva, Universidade Federal da Bahia, Salvador, Bahia, Brazil, 3 Faculty of Epidemiology and Population Health, London School of Hygiene & Tropical Medicine, London, United Kingdom, 4 Faculdade de Economia, Universidade Federal da Bahia, Salvador, Bahia, Brazil, 5 Fundação Getúlio Vargas, São Paulo, São Paulo, Brazil

‡ Co-senior authors
* lucianacardim@yahoo.com.br

## Abstract

### Background

People living in economically disadvantaged circumstances experience higher risks of infections and death from arboviruses. However, more evidence is needed to better understand the socioeconomic factors influencing dengue mortality. We investigated if people of lower socioeconomic conditions in Brazil are more likely to die following dengue infection.

### Methodology/Principal Findings

Linking nationwide socioeconomic data from the 100 Million Brazilian Cohort with dengue disease and death records registered in Brazil between 1st January 2007 and 31st December 2018, we used multivariable hierarchical analysis to investigate the socioeconomic determinants of dengue-specific and all-cause mortality within 15 days of dengue symptom onset. Among the 3,018,131 individuals from the 100 Million Brazilian Cohort diagnosed with dengue, 1810 died from dengue (Case Fatality Rate (CFR)=0.06%, 95%CI = 0.06-0.06%) and 3076 (CFR = 0.10%, 95%CI = 0.10-0.11%) died from any cause within 15 days of dengue symptom onset. People residing in the Northeast (OR=2.32; 95%CI = 1.74-3.10) and Midwest (OR=1.68; 95%CI = 1.25-2.27) regions, self-identifying as black race/ethnicity (OR=1.58; 95%CI = 1.31-1.90), having lower level of education (OR=2.35, 95%CI = 1.17-4.73), being retired/receiving pension (OR=2.24; 95%CI = 1.76-2.86), living in a household with rudimentary sewage (OR=1.19; 95%CI = 1.04-1.37) and having >2 inhabitants

**Data availability statement:** The unidentified data underlying this article can be accessed on reasonable request to CIDACS Fiocruz and after ethical approval. For more information: https://cidacs.bahia.fiocruz.br/.

**Funding:** This work was supported by the British Council Newton Fund (Grant number 527418645 to EBB). EPS is a Wellcome Trust fellow (225925/Z/22/Z). JMP is a Wellcome Trust fellow (305644/Z/23/Z). The funder had no role in study design, data collection and analysis, decision to publish, or preparation of the manuscript.

**Competing interests:** The authors have declared that no competing interests exist.

per room (OR=1.31; 95%CI = 1.11-1.55) had at higher odds of dengue-specific mortality. Similar characteristics were also associated with higher all-cause mortality after dengue infection, but also included residing in North region (OR=1.60; 95%CI = 1.24-2.06) and rural areas (OR=1.12; 95%CI = 1.01-1.24), self-identifying as Asian (OR=1.65; 95%CI = 1.07-2.54) and mixed race/brown (OR=1.20; 95%CI = 1.10-1.31) and living in households with poorer quality building and sanitary conditions.

## Conclusions/Significance

Our findings provide evidence that individuals in Brazil with lower socioeconomic condition experience increased odds of dengue-specific and all-cause mortality within 15 days of dengue symptom onset. These findings underscore the importance of ensuring equitable access to high-quality treatment for severe dengue and suggest that reducing poverty and social inequality, including through improvement of sanitation and housing, may help mitigate dengue-related mortality.

### Author summary

Dengue fever disproportionately affects individuals living in economically disadvantaged areas. Using socioeconomic data from the 100 Million Brazilian Cohort linked to compulsory notification records for dengue and death certificates, this study investigates the socioeconomic factors associated with death following dengue cases in Brazil. We analysed over 3 million dengue cases reported between 2007 and 2018 and found that individuals from poorer regions, such as the Northeast, self-identifying as black race/ethnicity, with lower levels of education, and living in poor housing conditions, face higher chances of dying from dengue or from any cause within 15 days of symptom onset. These findings underscore that lower socioeconomic condition likely exacerbates the outcomes of dengue infection. Although further research is needed to understand the causal mechanisms including the role of co-morbidities, the findings of this study indicate that addressing poverty and inequality, improving sanitation and living conditions, and ensuring equitable access to high-quality healthcare may help to contribute to reducing dengue-related deaths.

## Introduction

Dengue virus (DENV) is an arthropod-borne virus (arbovirus) primarily transmitted by anthropophilic *Aedes* mosquitoes. DENV belongs to the *Flaviviridae* family and has four distinct serotypes (DENV1–4) [1], which can immunologically cross-react with each other as well as the closely related Zika virus (ZIKV) (i.e., in the case of DENV2–4) [2]. Previous DENV infections have the potential to either provide immune protection or trigger antibody-dependent enhancement and increase the risk of developing severe disease and, consequently, death [3,4]. In the Americas, the case

fatality rate among severe dengue cases is estimated to be 1–2% [1], but this can be reduced to less than 1% under expert clinical management with careful fluid replacement [5]. Therefore, early detection and adequate management of severe dengue cases are considered key measures for mitigating the risk of dengue mortality [5].

Despite efforts to enhance vector control, dengue cases continued to rise in many countries, likely associated with high demographic density of the urban centre, resource constraints, and changes in climatic variables, such as precipitation, humidity, and temperature [6]. Additionally, dengue incidence and mortality remain disproportionately high in low- and middle-income countries [7]. In 2024, more than 14 million dengue cases and almost 10,000 dengue-related deaths were reported globally [8]. Of these, over 12 million cases occurred in the Americas, with an incidence of 1338 per 100,000 inhabitants, representing a 195% increase compared to 2023 and a 371% increase compared to the average in the previous five years [9].

A higher risk of DENV infection has been previously associated with lower education, living in poverty and households of poor housing materials or interrupted water supply, low healthcare coverage, being married, divorced or widowed, and non-white ethnicities in a 2022 systematic review looking at arbovirus infections in 36 studies from 23 countries [10]. In addition, a scoping review of 78 studies identified several social determinants associated with dengue mortality, including race/ethnicity, education, poverty, care-seeking behaviour, healthcare access, quality of care, and health staff knowledge [11].

In Brazil, DENV infections are similarly more frequent in people with lower socioeconomic status [12–14]. However, a 2002–2003 case-control study conducted in a large Brazilian capital (Salvador, Brazil) found people of white race/ethnicity to have higher odds of severe disease compared to people of black race/ethnicity [15]. Furthermore, most studies looking at socioeconomic drivers of mortality, including race, adjust for a multiplicity of factors that are in the causal pathway between race/ethnicity and health. To address this knowledge gap, we used a hierarchical approach that considers race/ethnicity a distal factor for dengue mortality that impacts access to education, housing and healthcare to investigate if individuals of lower socioeconomic conditions are at higher chance of death following dengue infection in Brazil, from 2007 to 2018.

## Materials and methods

### Ethics statement

This study was performed under the international (Helsinki), Brazilian and UK research regulations. Approval for this research was provided by the ethics committees from Instituto Gonçalo Muniz - Oswaldo Cruz Foundation (number 1 612 302 in 2016 and 4 756 567 in 2021) and from the London School of Hygiene & Tropical Medicine (number 25.339/2021).

Formal consent was not obtained, as the study used anonymized secondary data. The deidentified dataset was provided exclusively for this study and further data access requests must be submitted to Cidacs/Fiocruz subject to approval from Oswaldo Cruz Foundation ethical committee.

### Study design and data source

In this cohort study, we linked socioeconomic data from the 100 Million Brazilian Cohort [16] to records of dengue cases registered in the Brazilian Notifiable Diseases Information System (*Sistema de Informação de Agravos de Notificação – SINAN*) and death records from the Brazilian Mortality Information System (*Sistema de Informação sobre Mortalidade - SIM*) between 1st January 2007 and 31st December 2018.

The 100 Million Brazilian Cohort contains demographic, economic and social data from low-income families who applied for social assistance programs through the Brazilian Unified Registry for Social Programs (*Cadastro Único para Programas Sociais* - CadÚnico) between 2001 and 2018 [16]. To enrol in CadÚnico, the family must meet one of the following criteria: i. have a monthly income per person of up to half a minimum wage, or ii. have a total family income of three minimum wages [17]. From the cohort, we extracted sex, date of birth, race/ethnicity, location of the household,

region of the family home, level of education, employment, housing material, household water supply, sewage disposal system, waste collection and household density.

SINAN is the national notifiable disease registry in Brazil [18]. The SINAN-dengue registry includes records for all suspected dengue cases collected using standardised forms by health professionals. A suspected case of dengue is defined as an individual who lives or has travelled in the last 14 days to an area with either known transmission of DENV or established presence of *Ae. aegypti* and presents with fever, usually lasting between 2 and 7 days, and two or more of the following symptoms: nausea, vomiting, rash, myalgia, headache, retro-orbital pain, petechial or positive tourniquet test, and leukopenia [19]. After notification, suspected cases are investigated and classified as confirmed or ruled out. Cases are further classified by severity as "dengue without warning signs", "dengue with warning signs", or "severe dengue". From SINAN-dengue, we extracted information on the final classification of the case and severity (ruled out, dengue, dengue with warning signs, and severe dengue), date of first symptoms, age at first symptoms, evolution of the case (cure, death from dengue, death from other causes, and death under investigation) and date of death.

SIM is the national mortality information system in Brazil. The issuance of a death certificate by a physician is mandatory for all deaths that occur in the country [20]. From SIM, we extracted information on the causes of death using the International Classification of Diseases 10th revision (ICD-10) and date of death.

The linkage of dengue and death records with the 100 Million Brazilian Cohort was performed using a record linkage tool (CIDACS-RL) based on a similarity score specifically designed to conduct large-scale data linkage using administrative data from Brazil [21]. From each database, we used five attributes (name, mother's name, date of birth, gender, and municipality of residence) to perform the linkage. When there was more than one notification per person independently of the notification date, we selected the one that presented the best similarity score during the linkage. The accuracy of the linkages was calculated, and the linkage of the 100 Million Brazilian cohort with the dengue registry had 96% sensitivity and 94% specificity. For the linkage of the cohort with the death registry, the sensitivity and specificity ranged from 97% to 100% depending on the study year. Linkage procedures were conducted at the Center of Data and Knowledge Integration for Health (*Centro de Integração de Dados e Conhecimentos para Saúde* - CIDACS) at the Oswaldo Cruz Foundation (*Fundação Oswaldo Cruz* - FIOCRUZ) in a strict data protection environment and according to national and international ethical and legal regulations [22]. After linkage, the data were de-identified and provided to the researchers for use in a safe haven without access to the internet.

## Study participants and definitions

In this study, we included all confirmed and probable cases of dengue. Probable cases are those confirmed based on clinical and epidemiological criteria [23]. We excluded from this study individuals with death records prior to 2007, dengue cases classified as ruled out, inconclusive, or without final classification, and duplicate records. The two outcomes were: i. dengue-specific deaths that occurred within 15 days of dengue symptom onset and ii. all-cause deaths that occurred within 15 days of dengue symptom onset. We considered dengue-specific deaths all individuals with a dengue death registry recorded in the SINAN-dengue and/or with dengue coded as any of the causes of death in the SIM (ICD-10 code A90 or A91). We considered all-cause deaths individuals with a death registry recorded in the SINAN-dengue and/or those individuals linked with the SIM, regardless of the registered cause of death. We use a limit of 15 days for death because the average incubation period of the virus is 5–6 days, the period for the development of severe cases typically ranges from 3 to 7 days after the onset of symptoms, and death generally occurs within 12–24 hours thereafter [24]. Deaths that occurred after 15 days, as well as those with an underlying cause categorized as external (ICD-10 code V01 to Y98), were not considered as fatalities.

## Data analysis

For the two studied outcomes, we calculated the case fatality rate following dengue case by year and for each socioeconomic and demographic characteristic. Among all-cause deaths following dengue, we calculated the proportional mortality

(PM) as the proportion of each underlying cause of death divided by the total number of deaths occurring within 15 days of dengue symptom onset.

We used a hierarchical approach to group socioeconomic and demographic variables into two blocks representing distal and proximal variables. The distal level included variables that are considered proxies for access to health care and included: area of residence (region of Brazil and location of the household - urban vs. rural) and race/ethnicity (i.e., as a proxy for systemic and structural racism) [25]. The proximal level included indicators of socioeconomic conditions that are possibly influenced by distal variables and included: level of education, employment, and household conditions (i.e., housing material, household water supply, sewage disposal system, waste collection and household density). Because age and sex were considered a priori confounders, they were included in all models (Fig 1).

Using logistic regression, we generated a first model to investigate the Odds Ratio (OR) and 95% Confidence Intervals (95% CI) for the association between the distal variables and mortality (Model 1). The second model examined the association between the proximal variables and mortality and also included all variables from Model 1 that were associated with the outcome mortality at a p-value of <0.1. Both models were based on a complete case analysis (i.e., individuals with missing data on any of the distal or proximal covariates were excluded from the hierarchical approach).

Finally, we conducted a sensitivity analysis including only laboratory-confirmed dengue cases, using the same hierarchical analytical approach as in the main analysis. All data analysis was performed using Stata version 15.1.

## Results

Between 2007 and 2018, were registered in SINAN-dengue 3,018,131 as confirmed or probable dengue case from the 100 Million Brazilian Cohort. A total of 3076 individuals died from any cause within 15 days of the onset of the first symptoms of dengue; of these, 58.8% (1810/3076) were registered as dengue (Fig 2). Notably, using only the SIM

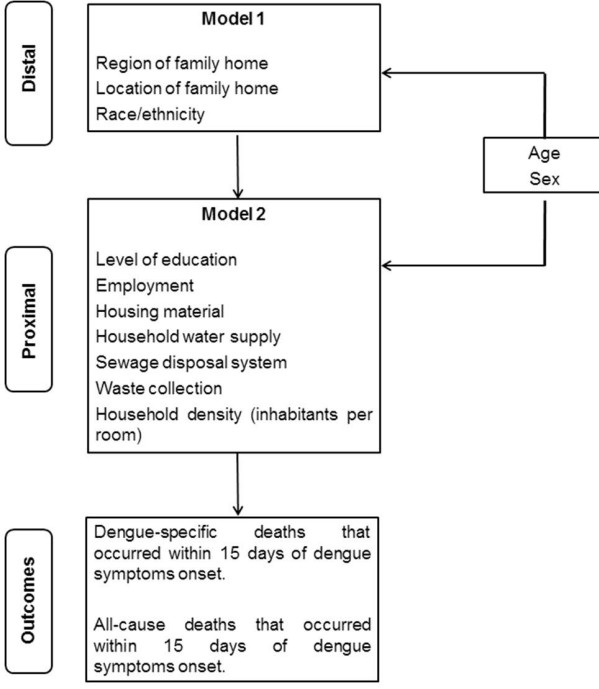

**Fig 1. Conceptual model of the relationship between socioeconomic and demographic characteristics grouped as Distal and Proximal variables and deaths among dengue cases in Brazil.** 2007-2018.

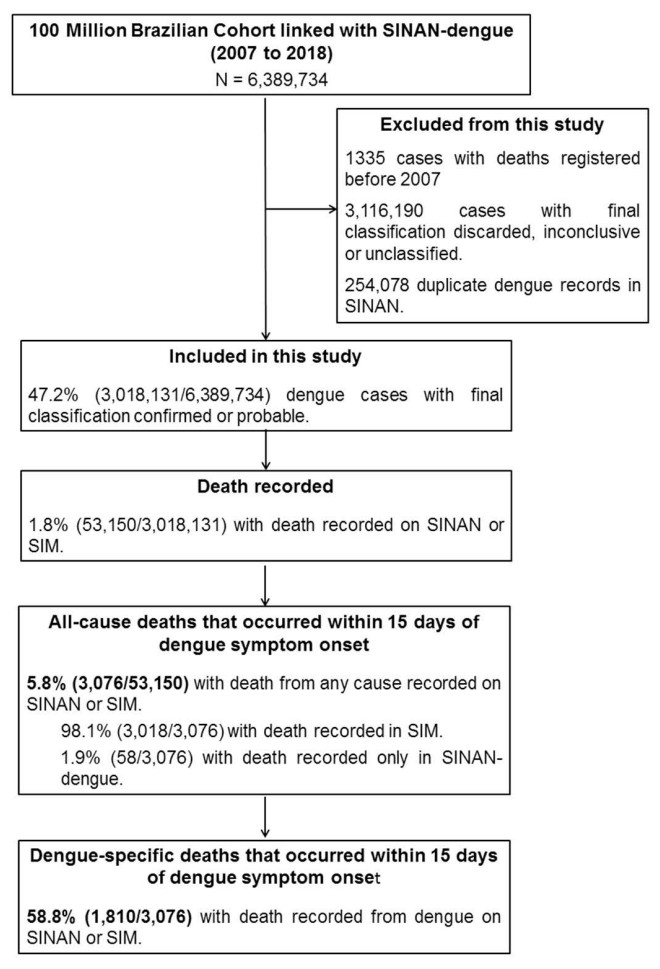

**Fig 2. Flowchart of study population.** Brazil. 2007-2018.

records, dengue and dengue haemorrhagic fever (ICD-10 code A90 and A91, respectively) were attributed as an underlying cause of death on only 38.1% (1149/3018) of the death records. The other top five underlying causes of death recorded in SIM included: unspecified pneumonia (123/3018; PM = 4.1%), ill-defined and unspecified causes of mortality (107/3018; PM = 3.5%), unspecified septicaemia (105/3018; PM = 3.5%), unspecified acute myocardial infarction (62/3018; PM = 2.1%), and unspecified bleeding disorder (35/3018; PM = 1.2%) (Table 1).

The dengue-specific case fatality rate was 0.06% (95%CI = 0.06-0.06%) and ranged from 0.05% (95%CI = 0.03-0.06%) in 2007 to 0.09% (95%CI = 0.08-0.11%) in 2008. The Northeast region had the highest dengue-specific case fatality rate, with 0.10% (95%CI = 0.09-0.10%) followed by Midwest, with 0.07% (95%CI = 0.07-0.08%). Higher dengue-specific case fatality rates within 15 days of dengue symptom onset were also observed in male (0.07%) individuals, aged ≥60 years (0.17%), self-identified as Asian (0.09%) or black (0.08%), lived in a rural area (0.07%), never attended school (0.10%), were retired or received pensions (0.25%) and resided in households built with coated mud, wood or other materials (0.07%), without public water network connections (0.08%), with rudimentary cesspool, ditch, or other outdoor sewage disposal (0.08%), without city waste collection (0.08%) and with more than two inhabitants per room (0.06%). The case fatality rate within 15

**Table 1. Ten most frequent underlying causes of death, except dengue and dengue haemorrhagic fever, within 15 days following the onset of symptoms of dengue. Brazil. 2007-2018.**

| Underlying cause of death | N | PM (%) |
|---|---|---|
| A91 - Dengue haemorrhagic fever | 636 | 21.1 |
| A90 - Dengue | 513 | 17.0 |
| J18.9 - Unspecified pneumonia | 123 | 4.1 |
| R99 - Other ill-defined and unspecified causes of mortality | 107 | 3.5 |
| A41.9 - Unspecified septicaemia | 105 | 3.5 |
| I21.9 - Unspecified acute myocardial infarction | 62 | 2.1 |
| D69.9 - Unspecified bleeding disorder | 35 | 1.2 |
| A09 - Diarrhea and gastroenteritis of presumed infectious origin | 33 | 1.1 |
| I64 - Stroke, not specified as hemorrhagic or ischemic | 32 | 1.1 |
| E14.9 - Unspecified diabetes mellitus - uncomplicated | 26 | 0.9 |
| K74.6 - Other and unspecified forms of liver cirrhosis | 26 | 0.9 |
| J18.0 - Unspecified pneumonia | 25 | 0.8 |
| Other types of causes | 1295 | 42.9 |
| Total* | 3018 | 100.0 |

PM: Proportional Mortality. Calculated as the ratio between the number of deaths due to this cause and the total number of death, multiplied by 100.

*58 individuals did not have information from underlying cause of death, as they were not linked with SIM. In SINAN, 77.6% (45/58) of these deaths were attributed to dengue, 12.1% (7/58) to other causes, and 10.3% (6/58) remained under investigation.

days of dengue symptom onset by all-causes was 0.10% (95%CI = 0.10-0.11%), with similar case fatality rate distributions across socioeconomic and demographic variables, as observed in dengue-specific deaths (Table 2).

Individuals aged 60 or over had higher odds compared to those aged 20–39 years for both dengue-specific (OR=3.74; 95%CI = 3.13-4.46) and all-cause deaths (OR=4.78; 95%CI = 4.20-5.44), within 15 days of dengue symptom onset. The association was stronger in males for dengue-specific deaths (OR=1.42; 95%CI = 1.28-1.58) and all-cause deaths (OR=1.56; 95%CI = 1.44-1.69). The distal variables of the region of household and race/ethnicity were associated with dengue-specific and all-cause deaths within 15 days of dengue symptom onset. Considering only dengue-specific deaths, the odds of death were elevated in individuals who lived in the Northeast (OR=2.32; 95%CI = 1.74-3.10) and Midwest (OR=1.68; 95%CI = 1.25-2.27) regions of Brazil, relative to those in the South region, and in individuals who self-identified as black (OR=1.58; 95%CI = 1.31-1.90), relative to those self-identifying as white. Generally similar estimates were observed for all-cause deaths within 15 days of dengue symptom onset; however, increased odds of all-cause deaths were also observed for individuals who lived in the North region (OR=1.60; 95%CI = 1.24-2.06) and in rural areas (OR=1.12; 95%CI = 1.01-1.24), and in individuals who self-identified as Asian (OR=1.65; 95%CI = 1.07-2.54) and mixed race/brown (OR=1.20; 95%CI = 1.10-1.31) (Table 3).

The analysis was performed by excluding individuals with missing data from all models. *Covariates in distal model, for both models, were adjusted for age and sex. **Covariates in proximal model for dengue-specific deaths were adjusted for covariates from distal model with p < 0.1 (i.e., region of family home and race/ethnicity), age and sex. ***Covariates in proximal model for all-cause deaths were adjusted for covariates from distal model with p < 0.1 (i.e., region of family home, location of the household and race/ethnicity), age and sex.

Proximal variables associated with dengue-specific deaths included education, employment status, sewage disposal and household density. The odds of dengue-specific deaths were elevated in individuals who attended preschool (OR=2.35, 95%CI = 1.17-4.73) and those who never attended school (OR=2.28, 95%CI = 1.21-4.32), compared with high

**Table 2. Baseline characteristics of individuals with dengue, number of dengue-specific deaths and all-cause deaths within 15 days following the onset of symptoms of dengue and case fatality rate. Brazil. 2007-2018.**

| Demographic variables | Number of dengue cases (%) | Number of dengue-specific deaths (%) | Case fatality rate from dengue (95% CI) | Number of all-cause deaths (%) | Case fatality rate from all cause (95% CI) |
|---|---|---|---|---|---|
| **Year** | | | | | |
| 2007 | 128,920 (4.3) | 58 (3.2) | 0.05 (0.03-0.06) | 111 (3.6) | 0.09 (0.07-0.10) |
| 2008 | 156,015 (5.2) | 145 (8.0) | 0.09 (0.08-0.11) | 202 (6.6) | 0.13 (0.11-0.15) |
| 2009 | 135,736 (4.5) | 97 (5.4) | 0.07 (0.06-0.09) | 149 (4.8) | 0.11 (0.09-0.13) |
| 2010 | 353,118 (11.7) | 191 (10.6) | 0.05 (0.05-0.06) | 293 (9.5) | 0.08 (0.07-0.09) |
| 2011 | 243,710 (8.1) | 185 (10.2) | 0.08 (0.07-0.09) | 262 (8.5) | 0.11 (0.10-0.12) |
| 2012 | 181,574 (6.0) | 121 (6.7) | 0.07 (0.06-0.08) | 204 (6.6) | 0.11 (0.10-0.13) |
| 2013 | 491,092 (16.3) | 228 (12.6) | 0.05 (0.04-0.05) | 352 (11.4) | 0.07 (0.06-0.08) |
| 2014 | 205,095 (6.8) | 158 (8.7) | 0.08 (0.07-0.09) | 237 (7.7) | 0.12 (0.10-0.13) |
| 2015 | 548,740 (18.2) | 292 (16.1) | 0.05 (0.05-0.06) | 502 (16.3) | 0.09 (0.08-0.10) |
| 2016 | 402,287 (13.3) | 184 (10.2) | 0.05 (0.04-0.05) | 444 (14.4) | 0.11 (0.10-0.12) |
| 2017 | 82,468 (2.7) | 73 (4.0) | 0.09 (0.07-0.11) | 158 (5.1) | 0.19 (0.16-0.22) |
| 2018 | 89,376 (3.0) | 78 (4.3) | 0.09 (0.07-0.11) | 162 (5.3) | 0.18 (0.15-0.21) |
| **Sex** | | | | | |
| Female | 1,780,094 (59.0) | 952 (52.6) | 0.05 (0.05-0.06) | 1543 (50.2) | 0.09 (0.08-0.09) |
| Male | 1,238,037 (41.0) | 858 (47.4) | 0.07 (0.07-0.07) | 1533 (49.8) | 0.12 (0.12-0.13) |
| **Age (years)** | | | | | |
| 0 - 4 | 112,873 (3.7) | 102 (5.6) | 0.09 (0.07-0.11) | 154 (5.0) | 0.14 (0.12-0.16) |
| 5 - 9 | 182,973 (6.1) | 157 (8.7) | 0.09 (0.07-0.10) | 181 (5.9) | 0.10 (0.09-0.11) |
| 10 - 19 | 689,102 (22.8) | 287 (15.9) | 0.04 (0.04-0.05) | 415 (13.5) | 0.06 (0.06-0.07) |
| 20 - 39 | 1,216,490 (40.3) | 478 (26.4) | 0.04 (0.04-0.04) | 773 (25.1) | 0.06 (0.06-0.07) |
| 40 - 59 | 618,929 (20.5) | 444 (24.5) | 0.07 (0.07-0.08) | 792 (25.8) | 0.13 (0.12-0.14) |
| >= 60 | 197,729 (6.6) | 342 (18.9) | 0.17 (0.16-0.19) | 761 (24.7) | 0.39 (0.36-0.41) |
| **Race/ethnicity** | | | | | |
| White | 979,814 (34.8) | 486 (29.0) | 0.05 (0.05-0.05) | 785 (27,6) | 0.08 (0.08-0.09) |
| Black | 186,681 (6.6) | 150 (9.0) | 0.08 (0.07-0.09) | 246 (8.6) | 0.13 (0.12-0.15) |
| Asian | 13,962 (0.5) | 13 (0.8) | 0.09 (0.00-0.16) | 21 (0.7) | 0.15 (0.09-0.23) |
| Mixed Brown | 1,626,384 (57.8) | 1021 (61.0) | 0.06 (0.06-0.07) | 1788 (62.8) | 0.11 (0.11-0.12) |
| Indigenous | 7,733 (0.3) | 4 (0.2) | 0.05 (0.01-0.13) | 9 (0.3) | 0.12 (0.05-0.22) |
| **Location of the household** | | | | | |
| Urban | 2,554,874 (86.8) | 1479 (84.0) | 0.06 (0.06-0.06) | 2480 (82.9) | 0.10 (0.09-0.10) |
| Rural | 389,358 (13.2) | 281 (16.0) | 0.07 (0.06-0.08) | 513 (17.1) | 0.13 (0.12-0.14) |
| **Region of family home** | | | | | |
| South | 133,753 (4.4) | 57 (3.2) | 0.04 (0.03-0.06) | 91 (3.0) | 0.07 (0.06-0.08) |
| Northeast | 810,862 (26.9) | 784 (43.4) | 0.10 (0.09-0.10) | 1309 (42.6) | 0.16 (0.15-0.17) |
| Southeast | 1,381,100 (45.8) | 517 (28.6) | 0.04 (0.03-0.04) | 905 (29.4) | 0.07 (0.06-0.07) |
| North | 236,690 (7.9) | 122 (6.8) | 0.05 (0.04-0.06) | 249 (8.1) | 0.11 (0.09-0.12) |
| Midwest | 453,331 (15.0) | 328 (18.1) | 0.07 (0.07-0.08) | 520 (16.9) | 0.12 (0.11-0.13) |
| **Level of education** | | | | | |
| University graduate | 50,225 (1.9) | 11 (0.7) | 0.02 (0.01-0.04) | 19 (0.7) | 0.04 (0.02-0.06) |
| Elementary and high school | 2,051,647 (78.2) | 1061 (67.1) | 0.05 (0.05-0.06) | 1805 (66.7) | 0.09 (0.08-0.09) |
| Pre-school | 77,992 (3.0) | 53 (3.4) | 0.07 (0.05-0.09) | 68 (2.5) | 0.09 (0.7-0.11) |
| Never went to school | 444,353 (16.9) | 457 (28.9) | 0.10 (0.09-0.11) | 813 (30.1) | 0.18 (0.17-0.20) |

*(Continued)*

**Table 2.** (Continued)

| Demographic variables | Number of dengue cases (%) | Number of dengue-specific deaths (%) | Case fatality rate from dengue (95% CI) | Number of all-cause deaths (%) | Case fatality rate from all cause (95% CI) |
|---|---|---|---|---|---|
| **Employment** | | | | | |
| Employed | 770,872 (28.4) | 423 (25.7) | 0.06 (0.05-0.06) | 790 (28.2) | 0.10 (0.10-0.11) |
| Unemployed | 1,901,165 (69.9) | 1106 (67.1) | 0.06 (0.06-0.06) | 1747 (62.3) | 0.09 (0.09-0.10) |
| Retired/pension | 47,296 (1.7) | 120 (7.3) | 0.25 (0.21-0.30) | 269 (9.6) | 0.57 (0.50-0.64) |
| **Housing material** | | | | | |
| Brick/masonry | 2,489,223 (85.6) | 1456 (83.4) | 0.06 (0.06-0.06) | 2413 (81.4) | 0.10 (0.09-0.10) |
| Coated mud, wood, others | 417,419 (14.4) | 290 (16.6) | 0.07 (0.06-0.08) | 551 (18.6) | 0.13 (0.12-0.14) |
| **Household water supply** | | | | | |
| Public network connection | 2,355,353 (81.0) | 1314 (75.3) | 0.06 (0.05-0.06) | 2192 (73.9) | 0.09 (0.09-0.10) |
| Water well, spring, others | 551,335 (19.0) | 432 (24.7) | 0.08 (0.07-0.09) | 772 (26.1) | 0.14 (0.13-0.15) |
| **Sewage disposal system** | | | | | |
| City public syst | 1,593,668 (55.1) | 781 (45.0) | 0.05 (0.05-0.05) | 1323 (44.9) | 0.08 (0.08-0.09) |
| Septic tank | 410,185 (14.2) | 268 (15.4) | 0.07 (0.06-0.07) | 454 (15.4) | 0.11 (0.10-0.12) |
| Rudimentary cesspool, ditch, others | 888,747 (30.7) | 687 (39.6) | 0.08 (0.07-0.08) | 1167 (39.6) | 0.13 (0.12-0.14) |
| **Waste collection** | | | | | |
| City collection | 2,491,960 (85.7) | 1423 (81.5) | 0.06 (0.05-0.06) | 2382 (80.4) | 010 (0.09-0.10) |
| No collection, burned, buried, others | 414,717 (14.3) | 323 (18.5) | 0.08 (0.07-0.09) | 582 (19.6) | 0.14 (0.13-0.15) |
| **Household density** | | | | | |
| ≤2 inhabitants per room | 532,636 (17.7) | 240 (13.3) | 0.05 (0.04-0.05) | 433 (14.1) | 0.08 (0.07-0.09) |
| >2 inhabitants per room | 2,485,495 (82.4) | 1570 (86.7) | 0.06 (0.06-0.07) | 2643 (85.9) | 0.11 (0.10-0.11) |

CI: Confidence Interval

[1]Individuals with missing data were included in the study population, but not in the adjusted analysis.

[2]Data is missing for the following variables: age - 35 (0.0%); race/ethnicity - 203,557 (6.7%); location of the household - 73,899 (2.5%); region of family home - 2,395 (0.1%); level of education - 393,914 (13.1%); employment - 298,798 (9.9%); housing material - 111,489 (3.7%); household water supply - 111,443 (3.7%); sewage disposal system - 125,531 (4.2%); waste collection - 111,454 (3.7%).

school education; individuals who were retired or receiving a pension (OR=2.24; 95%CI = 1.76-2.86), compared with employed; individuals living in households with a rudimentary cesspool, ditch or other outdoor sewage disposal (OR=1.19; 95%CI = 1.04-1.37), compared with public sewage; and individuals living in household with more than two inhabitants per room (OR=1.31; 95%CI = 1.11-1.55) compared to those that have two or fewer inhabitants per room. For all-cause deaths following dengue cases, increased odds of death were also higher among individuals who never attended school (OR=2.53; 95%CI = 1.56-4.13), compared with those with high school education; individuals who were retired or receiving pension (OR=2.31; 95%CI = 1.95-2.73), compared with employed; and, in general, individuals living in households with poorer quality building and sanitary conditions (Table 3).

When included only laboratory-confirmed dengue cases, we obtained similar point estimates to the main analysis which included both probable and confirmed cases. The direction and magnitude of the associations remained consistent across both models, except for household location, where the odds decreased for rural areas for dengue-specific deaths (OR=0.80; 95%CI = 0.68-0.93) and all-cause deaths (OR=0.83; 95%CI = 0.72-0.96) (S1 Table).

**Table 3. Multivariate hierarchical model used to assess socioeconomic factors associated with dengue-specific deaths and all-cause deaths within 15 days following the onset of symptoms of dengue. Brazil. 2007-2018.**

| Variables | Dengue-specific deaths | | All-cause deaths | |
|---|---|---|---|---|
| | P-value | OR crude (95% CI) | P-value | OR crude (95% CI) |
| **Age (years)** | | | | |
| 20 - 39 | | 1 | | 1 |
| 0 - 4 | <0.001 | 2.83 (2.10-3.82) | <0.001 | 2.13 (1.66-2.73) |
| 5 - 9 | <0.001 | 1.78 (1.41-2.25) | 0.142 | 1.16 (0.95-1.42) |
| 10 - 19 | 0.486 | 0.94 (0.79-1.12) | 0.007 | 0.82 (0.76-0.95) |
| 40 - 59 | <0.001 | 1.87 (1.60-2.18) | <0.001 | 2.08 (1.85-2.33) |
| >= 60 | <0.001 | 3.74 (3.13-4.46) | <0.001 | 4.78 (4.20-5.44) |
| **Sex** | | | | |
| Female | | 1 | | 1 |
| Male | <0.001 | 1.42 (1.28-1.58) | <0.001 | 1.56 (1.44-1.69) |
| **Distal variables** | **P-value** | **OR adjusted* (95% CI)** | **P-value** | **OR adjusted* (95% CI)** |
| **Region of family home** | | | | |
| South | | 1 | | 1 |
| Northeast | <0.001 | 2.32 (1.74-3.10) | <0.001 | 2.42 (1.93-3.04) |
| Southeast | 0.439 | 0.89 (0.67 -1.19) | 0.949 | 0.99 (0.79-1.24) |
| North | 0.149 | 1.28 (0.92-1.78) | <0.001 | 1.60 (1.24-2.06) |
| Midwest | 0.001 | 1.68 (1.25-2.27) | <0.001 | 1.64 (1.30-2.07) |
| **Location of the household** | | | | |
| Urban | | 1 | | 1 |
| Rural | 0.652 | 1.03 (0.90-1.18) | 0.027 | 1.12 (1.01-1.24) |
| **Race/ethnicity** | | | | |
| White | | 1 | | 1 |
| Black | <0.001 | 1.58 (1.31-1.90) | <0.001 | 1.55 (1.34-1.79) |
| Asian | 0.077 | 1.65 (0.95-2.86) | 0.024 | 1.65 (1.07-2.54) |
| Mixed Brown | 0.177 | 1.08 (0.97-1.21) | <0.001 | 1.20 (1.10-1.31) |
| Indigenous | 0.809 | 0.89 (0.33-2.38) | 0.596 | 1.20 (0.62-2.31) |
| **Proximal variables** | **P-value** | **OR adjusted** (95% CI)** | **P-value** | **OR adjusted*** (95% CI)** |
| **Level of education** | | | | |
| University graduate | | 1 | | 1 |
| Elementary and high school | 0.061 | 1.82 (0.97-3.40) | 0.018 | 1.79 (1.11-2.89) |
| Pre-school | 0.017 | 2.35 (1.17-4.73) | 0.003 | 2.31 (1.33-4.01) |
| Never went to school | 0.011 | 2.28 (1.21-4.32) | <0.001 | 2.53 (1.56-4.13) |
| **Employment** | | | | |
| Employed | | 1 | | 1 |
| Unemployed | 0.062 | 1.14 (0.99-1.31) | 0.174 | 1.07 (0.97-1.19) |
| Retired/pension | <0.001 | 2.24 (1.76-2.86) | <0.001 | 2.31 (1.95-2.73) |
| **Housing material** | | | | |
| Brick/masonry | | 1 | | 1 |
| Coated mud, wood, others | 0.586 | 1.04 (0.89-1.22) | 0.024 | 1.14 (1.02-1.29) |
| **Household water supply** | | | | |
| Public network connection | | 1 | | 1 |
| Water well, spring, others | 0.058 | 1.15 (1.00-1.34) | <0.001 | 1.25 (1.11-1.40) |
| **Sewage disposal system** | | | | |
| City public syst | | 1 | | 1 |

*(Continued)*

**Table 3.** (Continued)

| Variables | Dengue-specific deaths | | All-cause deaths | |
|---|---|---|---|---|
| | P-value | OR crude (95% CI) | P-value | OR crude (95% CI) |
| Septic tank | 0.540 | 1.05 (0.89-1.24) | 0.955 | 1.00 (0.88-1.14) |
| Rudimentary cesspool, ditch, others | 0.012 | 1.19 (1.04-1.37) | 0.012 | 1.15 (1.03-1.28) |
| **Waste collection** | | | | |
| City collection | | 1 | | 1 |
| No collection, burned, buried, others | 0.150 | 0.88 (0.74-1.05) | 0.494 | 1.05 (0.91-1.23) |
| **Household density** | | | | |
| ≤2 inhabitants per room | | 1 | | 1 |
| >2 inhabitants per room | 0.001 | 1.31 (1.11-1.55) | 0.002 | 1.22 (1.07-1.37) |
| OR: Odds Ratio | | | | |
| CI: Confidence Interval | | | | |

## Discussion

In this nationwide cohort study of more than three million dengue cases diagnosed among low-income individuals in Brazil, we found evidence that people within the lowest socioeconomic conditions experience increased odds of dengue-specific and all-cause deaths during the 15 days following dengue symptom onset. Individuals aged or older than 60 years, male, living in the Northeast region that has some of the most widespread poverty in the country, who self-identified as having black race/ethnicity, who had lower or no education, who were retired or receiving a pension, and/or who resided in unfavourable living conditions had elevated case fatality rates within 15 days of dengue symptom onset. We also observed poorly defined underlying causes of death in some certificates, suggesting possible diagnostic delays or deficiencies in clinical management. Additionally, some underlying causes of death frequently recorded in SIM, such as diabetes and cardiovascular diseases, were likely exacerbated by dengue infection rather than being isolated outcomes. Given the systemic impact of dengue, it is plausible that these conditions contributed to mortality wholly or in part due to the infection.

Prompt and appropriate healthcare for dengue is key to preventing deaths [5]. Brazil faces significant inequalities in access to healthcare services, reflecting stark regional and socioeconomic disparities. Evidence from the Brazilian National Health Survey indicates that the North, Northeast and Midwest regions of the country record the lowest proportion of people accessing medical appointments [26], which can lead to delays in timely diagnosis and treatment and consequently increase the chance of death after dengue onset. We observed higher odds of all-cause deaths during the 15 days following dengue symptom onset among people living in rural rather than urban areas. This observation aligns with a case-control study conducted in Brazil that found individuals with severe dengue living in rural areas were almost 3-times more likely to die than those living in urban areas [27], which could be attributed to a combination of low socio-economic status in rural areas and limited access to healthcare services, with a lack of timely diagnosis and appropriate treatment [27,28]. However, in the analysis restricted to laboratory-confirmed dengue cases, rural residence was associated with a lower odds of mortality, diverging from the main analysis. One possible explanation is selection bias due to differential access to laboratory testing by household location. The cost of diagnostic tests is high [29], and in rural areas, testing is often limited and primarily available for severe or hospitalized cases. This may lead to the underrepresentation of laboratory confirmed milder dengue cases in rural areas, potentially producing a spurious protective association.

Although a case-control study conducted in Brazil has reported that individuals self-identifying as having Afro-Brazilian ethnicity and African ancestry had reduced risk of dengue haemorrhagic fever, which is on the causal pathway to dengue death [15] our findings indicate that individuals self-identifying as black or mixed brown race/ethnicity had up to 1.6-times

higher odds of death within 15 days of dengue symptom onset than those who self-identified as white. This finding seems to be contradictory, however this higher odds of death is possibly influenced by social issues such as unequal exposure to the risk of dengue infection, difficulty in accessing health services and the quality and speed of care received, which is essential for preventing dengue mortality. In Brazil, structural racism manifests as a social determinant of health, reflecting racialized disparities in factors including environmental sanitation and access to the Unified Health System (*Sistema Único de Saúde* - SUS) and results in health and social inequalities among the black population [30,31]. Within this context, the higher odds of death in individuals self-identifying as black or mixed brown race/ethnicity are particularly relevant, as they represent the largest segment of the Brazilian population and illustrate how racial and socioeconomic inequalities intersect to increase vulnerability. We also observed increased all-cause mortality among individuals self-identifying as Asian. Given the small size and heterogeneity of this group in Brazil, and the lack of previous evidence, this result should be interpreted with caution and warrants further investigation.

In our study, the odds of death were twice as high among individuals with lower or no educational attainment compared to those with high school education. Our results are similar to a population-based case-control study carried out in Brazil that observed that individuals who had less than four years of schooling had higher odds of dying from severe dengue (OR=1.44; 95%CI = 1.12-1.84) [27]. Education may be linked to dengue mortality in several ways. Limited awareness or understanding of dengue, particularly its progression to severe cases, has been shown to be a significant determinant of mortality [27,32]. Moreover, level of education serves as a marker of socioeconomic position, reflecting income disparities that can influence access to healthcare throughout a person's life [33,34].

Several factors may contribute to the increased dengue mortality in older adults, including a high prevalence of co-morbidities such as diabetes, hypertension and cardiovascular diseases [35–37], as well as diminished physiological resilience and immune function that accompanies ageing [38]. People who are retired or living out of pensions had higher odds of death even after adjusting for age, suggesting that retirement status may capture other dimensions of vulnerability beyond chronological aging. These may include limited social support, or barriers to accessing timely healthcare [39]. Additionally, in Brazil, being retired or on a pension is often linked to socioeconomic disadvantage, particularly among those receiving minimum-benefit pensions. These factors may contribute to delayed recognition of severity and reduced capacity to seek or receive prompt medical care.

Finally, we found that people living in poor housing conditions have a higher odds of mortality among dengue cases. Poor housing conditions, such as living in overcrowded areas, informal settlements, or slums with limited access to basic health and sanitation services, can create favourable environments for vector replication and biting, increasing the risk of arbovirus infections [40]. In addition, it is important to consider the possibility of a higher effect of antibody-dependent enhancement in people more frequently exposed to infections, in which successive heterotypic DENV infections may increase the risk of developing severe dengue and dengue mortality [41]. Furthermore, the mechanisms by which social status can negatively affect health are diverse and could be related to a range of factors associated with low socioeconomic condition including malnutrition and difficulty purchasing food, barriers to accessing health and social services, waiting periods for receiving social assistance, limited resources to adopt a healthy lifestyle behaviours, and stress [34,42].

Our study is subject to some limitations. Mainly, our linked database includes only one dengue notification per individual, the record with the highest similarity linkage score. As a result, we were unable to capture multiple infections in the same person or to identify which infection was the most severe. Although this may have led to an underestimation of dengue mortality within the cohort, it is unlikely that the selection of the notification was differential according to individuals' demographic or socioeconomic characteristics, and therefore unlikely to have biased our estimates. Second, as this study was conducted using secondary data, we have missing data in key variables (i.e., up to 13% of missing in the variable level of education and 10% of missing in the variable for employment), which were excluded from the adjusted analysis. However, performing a Missing Indicator Analysis was not possible as all missing indicators were omitted from the final model due to the small percentages. Third, we were unable to control for others determinants of dengue mortality,

such as comorbidities and pregnancy [43–45] as these data were not available in our database. Nonetheless, we consider comorbidities to mediate the association between social disadvantage to dengue mortality, since these conditions are more prevalent among socially disadvantage population [46]. In addition, it is important to highlight that our analysis were restricted to individuals included in CadÚnico, who already represent a population with marked socioeconomic vulnerability. As a result, the differences in the case fatality rate observed may underestimate the true magnitude of inequalities in dengue outcomes. If the entire Brazilian population and its broader spectrum of living conditions had been considered, disparities in case fatality rate would likely be even more pronounced. Therefore, we believe that this limitation does not weaken the interpretation of the association observed.

An additional limitation relates to diagnostic classification. Our analysis included both laboratory-confirmed and probable dengue cases, based on national surveillance definitions [23]. While this approach improves representativeness, it may introduce misclassification, as some probable cases may not be dengue infections. This non-differential misclassification could attenuate the observed associations. However, results from the sensitivity analysis restricted to laboratory-confirmed cases were largely consistent in direction and magnitude, supporting the robustness of our main findings.

Despite these limitations, by linking this large number of dengue cases with death registries and detailed socioeconomic information, we were able to investigate the link between proxies of poverty and deaths among confirmed and probable cases of dengue among the poorest half of the Brazilian population, i.e., those applying for social programmes in CadÚnico and enrolled in the 100 Million Brazilian Cohort baseline. We were also able to capture dengue deaths that were not linked to SIM and provide more accurate estimates of dengue case fatality in Brazil. Finally, by analysing all-cause deaths occurring within 15 days of dengue symptom onset, we were able to identify cases likely linked to dengue virus infection that were not explicitly listed as cause of death on Death Certificates. This approach acknowledges the well-established role of certain comorbidities in increasing the risk of mortality from dengue [47,48]. Thus, we recommend conducting matched cohort studies and self-controlled case series to better understand whether the deaths are primarily attributable to DENV infections or if they are disproportionately elevated due to lower socioeconomic conditions.

Our study showed that disparities in dengue-related deaths are strongly linked to individual and family social inequalities. Our findings highlight the urgent need for policies that mitigate dengue-related mortality, particularly by addressing health inequalities. To achieve this, specific measures are needed to promote equitable healthcare access and improve outcomes for vulnerable populations. In Brazil, expanding the coverage and strengthening the capacity of the Family Health Strategy (*Estratégia Saúde da Família* – ESF) is essential. Family doctors and primary care teams should be trained to recognize early signs of severe dengue and ensure timely referral to specialized care. Local governments, under the coordination of the Ministry of Health and its Secretariat of Health Surveillance, are responsible for implementing these actions. This includes ensuring equitable access to high-quality treatment for severe dengue and guaranteeing healthcare access for historically marginalized groups in Brazil. Additionally, expanding coverage of new dengue vaccines for individuals at higher odds of death should be a key consideration.

## Supporting information

**S1 Table. Sensitivity analysis including only laboratory-confirmed dengue cases to assess socioeconomic factors associated with dengue-specific deaths and all-cause deaths within 15 days following the onset of symptoms of dengue.** Brazil. 2007–2018.
(DOCX)

## Acknowledgments

We thank the data production team CIDACS/FIOCRUZ collaborators for their work linking these data and for providing information on data quality.

## Author contributions

**Conceptualization:** Luciana Lobato Cardim, Maria da Glória Teixeira, Maria da Conceição N. Costa, Elizabeth B. Brickley, Enny S. Paixão, Julia M. Pescarini.

**Data curation:** Luciana Lobato Cardim.

**Formal analysis:** Luciana Lobato Cardim.

**Funding acquisition:** Elizabeth B. Brickley, Enny S. Paixão, Julia M. Pescarini.

**Methodology:** Luciana Lobato Cardim, Maria da Glória Teixeira, Enny S. Paixão, Julia M. Pescarini.

**Project administration:** Elizabeth B. Brickley, Enny S. Paixão, Julia M. Pescarini.

**Supervision:** Liam Smeeth, Mauricio L. Barreto, Elizabeth B. Brickley, Enny S. Paixão, Julia M. Pescarini.

**Validation:** Luciana Lobato Cardim, Maria da Glória Teixeira, Maria da Conceição N. Costa, Wema Meranda Mitka, Camila Silveira Silva Teixeira, Andreia C. Santos, Gervasio Santos, André Portela Fernandes de Souza, Liam Smeeth, Mauricio L. Barreto, Elizabeth B. Brickley, Enny S. Paixão, Julia M. Pescarini.

**Visualization:** Luciana Lobato Cardim, Maria da Glória Teixeira, Maria da Conceição N. Costa, Wema Meranda Mitka, Camila Silveira Silva Teixeira, Andreia C. Santos, Gervasio Santos, André Portela Fernandes de Souza, Liam Smeeth, Mauricio L. Barreto, Elizabeth B. Brickley, Enny S. Paixão, Julia M. Pescarini.

**Writing – original draft:** Luciana Lobato Cardim, Wema Meranda Mitka, Enny S. Paixão, Julia M. Pescarini.

**Writing – review & editing:** Luciana Lobato Cardim, Maria da Glória Teixeira, Maria da Conceição N. Costa, Wema Meranda Mitka, Camila Silveira Silva Teixeira, Andreia C. Santos, Gervasio Santos, André Portela Fernandes de Souza, Liam Smeeth, Mauricio L. Barreto, Elizabeth B. Brickley, Enny S. Paixão, Julia M. Pescarini.

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
