## [Decision Letter · Decision Letter 0]

15 Jun 2025

PNTD-D-25-00509

Socioeconomic risk markers of dengue mortality in the 100 Million Brazilian Cohort (2007-2018): A nationwide registry-based cohort study

Dear Dr. Cardim,

Thank you for submitting your manuscript to PLOS Neglected Tropical Diseases. After careful consideration, we feel that it has merit but does not fully meet PLOS Neglected Tropical Diseases's publication criteria as it currently stands. Therefore, we invite you to submit a revised version of the manuscript that addresses the points raised during the review process.

Please submit your revised manuscript within 60 days, August 10, 2025. If you will need more time than this to complete your revisions, please reply to this message or contact the journal office at plosntds@plos.org. Please include the following items when submitting your revised manuscript:

We look forward to receiving your revised manuscript.

Kind regards,

Md. Kamrujjaman, Ph.D

Academic Editor

Qu Cheng

Section Editor

Shaden Kamhawi

co-Editor-in-Chief

Paul Brindley

co-Editor-in-Chief

**Comments to the Authors:**

**Please note that one of the reviews is uploaded as an attachment.**

**Journal Requirements:**

At this stage, the following Authors/Authors require contributions: Luciana Lobato Cardim, Maria da Glória Teixeira, Maria da Conceição N. Costa, Wema Meranda Mitka, Camila Silveira Silva Teixeira, Andreia C. Santos, Gervasio Santos, André Portela Fernandes de Souza, Liam Smeeth, Mauricio L. Barreto, Elizabeth B. Brickley, Enny S. Paixão, and Julia M. Pescarini. Please ensure that the full contributions of each author are acknowledged in the "Add/Edit/Remove Authors" section of our submission form.

2) We have noticed that you have uploaded Supporting Information files, but you have not included a list of legends. Please add a full list of legends for your Supporting Information files after the references list.

3) We note that you have indicated that there are restrictions to data sharing for this study. PLOS only allows data to be available upon request if there are legal or ethical restrictions on sharing data publicly. For more information on unacceptable data access restrictions, please see https://journals.plos.org/plosntds/s/data-availability#loc-unacceptable-data-access-restrictions.

b) If there are no restrictions, please upload the minimal anonymized data set necessary to replicate your study findings to a stable, public repository and provide us with the relevant URLs, DOIs, or accession numbers. For a list of recommended repositories, please see https://journals.plos.org/plosone/s/recommended-repositories. You also have the option of uploading the data as Supporting Information files, but we would recommend depositing data directly to a data repository if possible.

**Reviewers' Comments:**

Reviewer's Responses to Questions

**Key Review Criteria Required for Acceptance?**

**Methods**

-Are the objectives of the study clearly articulated with a clear testable hypothesis stated?

-Is the study design appropriate to address the stated objectives?

-Is the population clearly described and appropriate for the hypothesis being tested?

-Is the sample size sufficient to ensure adequate power to address the hypothesis being tested?

-Were correct statistical analysis used to support conclusions?

-Are there concerns about ethical or regulatory requirements being met?

Reviewer #1: (No Response)

Reviewer #2: This is a study of great social relevance, with detailed and clearly explained methodology, as well as a consistent discussion.

Reviewer #3: (No Response)

**Results**

-Does the analysis presented match the analysis plan?

-Are the results clearly and completely presented?

-Are the figures (Tables, Images) of sufficient quality for clarity?

Reviewer #1: (No Response)

Reviewer #2: The results are consistent and presented appropriately, I only suggest reviewing table 2.

Review the race/ethnicity variable number “1.021” in Table 2, as well as the employment variable number “1.106”

Reviewer #3: (No Response)

**Conclusions**

-Are the conclusions supported by the data presented?

-Are the limitations of analysis clearly described?

-Do the authors discuss how these data can be helpful to advance our understanding of the topic under study?

-Is public health relevance addressed?

Reviewer #1: (No Response)

Reviewer #2: It presented results relevant to public health, consistent with the objectives and method used. The limitations are important, however, they do not compromise the result.

Reviewer #3: (No Response)

**Editorial and Data Presentation Modifications?**

Reviewer #1: (No Response)

Reviewer #2: The hierarchical analysis made it possible to identify how the different levels interacted, affecting the health of the most vulnerable individuals. In this regard, we saw in the results that black and Asian individuals are more likely to die from dengue fever. In the discussion, this finding was attributed to structural racism and social conditions that affect black or mixed-brown people. And as for the Asian group, would it also be due to conditions similar to those of the other groups? Why did you choose not to associate the black group with the mixed-brown group?

Reviewer #3: (No Response)

**Summary and General Comments**

Reviewer #1: Using socioeconomic data from the 100 Million Brazilian Cohort linked to compulsory notification records for dengue and death certificates, this cohort study investigates the socioeconomic factors associated with death following dengue cases in Brazil. Results suggest that people within the lowest socioeconomic position experience increased risks of dengue-specific and all-cause deaths during the 15 days following dengue symptom onset. The findings emphasize the profound significance of rational allocation of healthcare resources in reducing the disease burden of dengue. However, I still have the following concerns:

Introduction. Please ensure all epidemiological data are properly referenced and verifiable, such as ‘In 2024, more than 14 million dengue cases and over 10,000 dengue-related deaths were reported globally.’

Methods. Please describe in detail the exclusion criteria for this study population and whether non-endemic areas, short-term residence, and immunized persons were considered.

Methods. The study outcome included all-cause deaths that occurred within 15 days of the onset of dengue symptoms, whether this included accidents. How the authors considered these causes of death.

Results. Since the authors viewed race/ethnicity as a distal factor in dengue mortality, affecting access to education, housing, and health care, which in turn is associated with dengue. This seems to be a framework for mediation analysis, and I'm curious about the results of the mediation analysis.

Results. Sensitivity analysis, such as multiple interpolation, is recommended to improve the reliability of the results.

Discussion. Elevated mortality risks for Black, Asian, and mixed-race individuals align with structural inequities, but the absence of adjustment for comorbidities (e.g., diabetes, hypertension) weakens causal interpretations. The authors acknowledge this limitation but should discuss its potential impact on OR estimates.

Discussion. It is recommended that the authors provide further discussion on the elevated case fatality rates within 15 days of dengue symptom onset for people who are retired or living on pensions. The authors mention that even after adjusting for age, the mortality rate is higher among those who are retired or living on pensions. This may have suggested that the effect of being retired or living on a pension is independent of aging. I think this is a novel phenomenon.

Discussion. The direction of the effect of place of residence when only laboratory-confirmed dengue cases are included contradicts the main analysis and requires further discussion.

Discussion. Please propose specific measures for equitable access to health care and health improvement, and state who is responsible for them. Family doctors or the CDC?

Inconsistent formatting (e.g., “OR=1.58” vs. “OR = 1.58”) should be unified.

Reviewer #2: The manuscript is clear and presents the results in a simple and direct way, facilitating understanding.

Reviewer #3: (No Response)

PLOS authors have the option to publish the peer review history of their article (what does this mean?). If published, this will include your full peer review and any attached files.

Reviewer #1: **Yes: **Yongfu Yu, School of Public Health, Fudan University, China

Reviewer #2: No

Reviewer #3: No

**Figure resubmission:**
---

## [Decision Letter · Decision Letter 1]

25 Sep 2025

PNTD-D-25-00509R1

Socioeconomic markers of dengue mortality in the 100 Million Brazilian Cohort (2007-2018): A nationwide registry-based cohort study

Dear Dr. Cardim,

Thank you for submitting your manuscript to PLOS Neglected Tropical Diseases. After careful consideration, we feel that it has merit but does not fully meet PLOS Neglected Tropical Diseases's publication criteria as it currently stands. Therefore, we invite you to submit a revised version of the manuscript that addresses the points raised during the review process.

Please submit your revised manuscript within 30 days Oct 25 2025. If you will need more time than this to complete your revisions, please reply to this message or contact the journal office at plosntds@plos.org. Please include the following items when submitting your revised manuscript:

We look forward to receiving your revised manuscript.

Kind regards,

Md. Kamrujjaman, Ph.D

Academic Editor

Qu Cheng

Section Editor

Shaden Kamhawi

co-Editor-in-Chief

Paul Brindley

co-Editor-in-Chief

**Additional Editor Comments:**

Reviewer #1:

Reviewer #3:

Reviewer #4:

**Journal Requirements:**

**Reviewers' comments:**

Reviewer's Responses to Questions

**Key Review Criteria Required for Acceptance?**

**Methods**

-Are the objectives of the study clearly articulated with a clear testable hypothesis stated?

-Is the study design appropriate to address the stated objectives?

-Is the population clearly described and appropriate for the hypothesis being tested?

-Is the sample size sufficient to ensure adequate power to address the hypothesis being tested?

-Were correct statistical analysis used to support conclusions?

-Are there concerns about ethical or regulatory requirements being met?

Reviewer #1: (No Response)

Reviewer #3: (No Response)

Reviewer #4: The study objectives are clearly stated with a testable hypothesis. The design, population, and sample size are appropriate, and the statistical analyses adequately support the conclusions. Ethical and regulatory requirements appear to have been met.

**Results**

-Does the analysis presented match the analysis plan?

-Are the results clearly and completely presented?

-Are the figures (Tables, Images) of sufficient quality for clarity?

Reviewer #1: (No Response)

Reviewer #3: (No Response)

Reviewer #4: The study adequately addresses all the points raised. The analysis presented aligns with the proposed analysis plan. The results are clearly and comprehensively presented, and the tables are of sufficient quality to ensure clarity and facilitate understanding. There are only two minor comments that the authors may consider as suggestions.

1. Page 11/line 246: In the Results section, in the sentence “…using only the SIM records, dengue and dengue haemorrhagic fever were attributed as an underlying cause of death…”, the authors refer to the older classification of dengue cases. I recommend explicitly noting that, although dengue notifications now follow the revised clinical classification, deaths in the Mortality Information System (SIM) are still coded using the older ICD-10 categories (A90 and A91), which do not distinguish cases with warning signs from severe dengue. This clarification would help readers understand that analyses based on SIM data reflect the older classification system.

2. Page 15/line 289/290: Looking at the table 3, the variables living in rural areas as well as self-identifying as Asian does not seems to be statistically significant.

**Conclusions**

-Are the conclusions supported by the data presented?

-Are the limitations of analysis clearly described?

-Do the authors discuss how these data can be helpful to advance our understanding of the topic under study?

-Is public health relevance addressed?

Reviewer #1: (No Response)

Reviewer #3: (No Response)

Reviewer #4: The conclusions of the manuscript are well supported by the data presented. The limitations of the analysis are clearly described, and the authors provide a thoughtful discussion of how these findings contribute to advancing our understanding of the socioeconomic factors associated with dengue mortality. The public health relevance of the results is appropriately addressed. Overall, the manuscript is well-structured and informative. There are three suggestions that the authors may consider when making adjustments:

1. Page 19/line 365: The authors draw attention to a purportedly increased all-cause mortality among individuals self-identified as Asian. It would be important to reference additional studies that have reported similar findings. Although this result appears atypical, it is borderline in terms of statistical significance (with the confidence interval very close to one) and, when weighed against the relevance of the other findings, does not seem to warrant particular emphasis. In this regard, it may be more relevant to highlight the higher all-causes mortality among individuals identifying as Brown (Pardo) compared to those identifying as White, given that Brown/mixed-race individuals constitute the largest group within the Brazilian population.

2. Page 21/Line 406: In this section, the authors state that it was not possible to determine the number of infections per individual, nor which episode was the most severe. This statement is not entirely clear. Would it not be possible to identify, at least within the study period, those cases with other reported episodes of dengue (notified as probable dengue cases) in the SINAN database?

3. The authors used CadÚnico data to classify cases according to social, economic, and household characteristics. This group is already highly vulnerable from a socioeconomic perspective. This suggests that, if the entire Brazilian population and its wide inequalities in living conditions were considered, differences in lethality would likely be much more pronounced. It would be appropriate for the authors to include a comment on this effect in their discussion.

**Editorial and Data Presentation Modifications?**

Reviewer #1: (No Response)

Reviewer #3: (No Response)

Reviewer #4: (No Response)

**Summary and General Comments**

Reviewer #1: The author has addressed most of the concerns. I think the current version of the manuscript is suitable for publication.

Reviewer #3: (No Response)

Reviewer #4: This article makes an important contribution to the discussion of socioeconomic factors associated with dengue mortality, through a novel analysis of a specific Brazilian population benefiting from income transfer programs. The methods employed are appropriate for addressing the research questions.

PLOS authors have the option to publish the peer review history of their article (what does this mean?). If published, this will include your full peer review and any attached files.

Reviewer #1: No

Reviewer #3: No

Reviewer #4: **Yes: **Gerusa Gibson

**Figure resubmission:**
---

## [Editor Report · Decision Letter 2]

19 Oct 2025

Dear Dr Cardim,

We are pleased to inform you that your manuscript 'Socioeconomic markers of dengue mortality in the 100 Million Brazilian Cohort (2007-2018): A nationwide registry-based cohort study' has been provisionally accepted for publication in PLOS Neglected Tropical Diseases.

Best regards,

Md. Kamrujjaman, Ph.D

Academic Editor

Qu Cheng

Section Editor

Shaden Kamhawi

co-Editor-in-Chief

Paul Brindley

co-Editor-in-Chief

---

## [Editor Report · Acceptance letter]

Dear Dr Cardim,

We are delighted to inform you that your manuscript, "Socioeconomic markers of dengue mortality in the 100 Million Brazilian Cohort (2007-2018): A nationwide registry-based cohort study," has been formally accepted for publication in PLOS Neglected Tropical Diseases.

Best regards,

Shaden Kamhawi

co-Editor-in-Chief

Paul Brindley

co-Editor-in-Chief
